# The Diversity and Composition of Insect Communities in Urban Forest Fragments near Panama City

**DOI:** 10.3390/biology14060721

**Published:** 2025-06-18

**Authors:** Jeancarlos Abrego, Enrique Medianero

**Affiliations:** 1Departamento de Zoología, Escuela de Biología, Facultad de Ciencias Naturales, Exactas y Tecnología, Universidad de Panamá, Ciudad de Panamá 3366, Panama; jean.abrego-l@up.ac.pa; 2Miembro del Sistema Nacional de Investigación (SNI-SENACYT), Ciudad de Panamá 0816-02852, Panama; 3Departamento de Ciencias Ambientales, Facultad de Ciencias Naturales, Exactas y Tecnología, Universidad de Panamá, Ciudad de Panamá 3366, Panama

**Keywords:** insects, alpha and beta diversity, morphospecies, urban forests, entomological communities, Panama

## Abstract

In Panama City, there are still small patches of forest that, despite being surrounded by buildings and roads, continue to host a wide variety of insects. In this study, we explored how diverse these insect communities are in four urban green areas. Even though the sites are close to each other and share similar conditions, we found that each one has a unique mix of species. We also observed that the larger and better-connected forest patches have more insect diversity, while the more isolated ones show fewer species. This suggests that not all green spaces in the city function the same way and that their size, location, and connectivity matter. Our results show that these urban forest fragments play a key role in supporting biodiversity and could serve as bridges or corridors for wildlife. Protecting and restoring them should be part of any plan to build a greener, healthier city.

## 1. Introduction

Panama City is a notable region in the neotropical region of America, given that it has conserved considerable natural forest coverage both in its interior and in the surrounding areas [1]. However, these green areas are not exempt from the effects of urban growth, including the division of the landscape into fragments of isolated vegetation [2].

Urbanization is a transformative factor in landscapes that can alter ecological and functional patterns at various scales [3,4,5]. Through mechanisms such as the modification of microclimates, changes in the composition of the soil, and the interruption of ecological flows, the urban environment can influence the distribution and composition of biological communities [6].

The reduction in extensive forest areas in isolated patches has been associated with variations in the richness, structure, and abundance of species in different taxonomic groups [7,8,9]. However, the magnitude and direction of these changes depend largely on the ecological context and the group of organisms under investigation [7].

Recent studies have emphasized the importance of understanding insect population dynamics in urban ecosystems, particularly in response to landscape structure, habitat restoration, and environmental stressors. For example, ref. [10] demonstrated that urban form and vegetation connectivity significantly shape insect community composition, even in highly developed areas. Similarly, ref. [11] found that green corridors and landscape heterogeneity enhance urban insect diversity, while ref. [12] highlighted how ecological restoration can reestablish functional insect communities. Moreover, ref. [13] reported that exposure to environmental contaminants alters insect assemblages in urban matrices, affecting their ecological roles. These findings underscore the need for studies in tropical cities to evaluate insect responses across fragmented urban landscapes and contribute to global knowledge on urban biodiversity resilience.

In recent decades, the need to include urban environments as valid settings for ecological research has been recognized, given the increasing expansion of such environments and their relevance with respect to efforts to conserve biodiversity [14,15,16]. Studies on urban diversity can facilitate not only the documentation of emerging biological patterns but also the evaluation of the roles of green areas as refuges or ecological corridors in landscapes that have been subjected to high levels of intervention [17,18,19].

This study aimed to estimate the diversity, abundance, and taxonomic composition of insects that belong to the families Ichneumonidae, Bethylidae, Asilidae, Dolichopodidae, Curculionidae, Membracidae, and Psocidae in four urban forest fragments that are located near Panama City. These seven families were selected due to their ecological representativeness, high abundance in tropical urban habitats, and relatively stable taxonomy at the subfamily and genus levels, which facilitates their morphological identification. In addition, they are considered functionally relevant groups due to their roles in pollination, predation, or parasitism, making them useful indicators of ecological integrity in fragmented landscapes. We sought to answer the following questions: (1) What are the richness, abundance, and composition of insects among the fragments under evaluation? (2) What level of similarity is evident among the communities associated with the four fragments under investigation?

Given that the fragments investigated in this research are geographically close to one another, similar in terms of altitude, climate, and type of vegetation, and located close to the Panama Canal, for many years, these fragments have maintained some degree of protection; accordingly, we hope to observe a high level of similarity among their entomological communities.

## 2. Materials and Methods

### 2.1. Study Area

This study was conducted in four urban forest fragments near Panama City: Metropolitan Natural Park (PNM, Los Momótides Trail), Albrook (ALB), Corozal (COR), and City of Knowledge (Cerro Gun) (Figure 1). These forest fragments are part of a biological corridor that runs along the eastern bank of the Panama Canal [20]. The forests of the Panama Canal Basin are located within protected areas and feature a rainy season that generally extends from May to November and a dry season that extends from December to April. These forests are characterized by different types of vegetation, but on the Pacific slope, which is close to Panama City, dry forest predominates [21]. The four sites under investigation are located within the tropical humid forest life zone [22].

In this study, we used the definition of fragments provided in [23], which defined a fragment as any patch of native vegetation around which most of the original vegetation has been removed. These fragments feature different types of soil and vegetation and vary in terms of size, shape, and isolation. Accordingly, conserved places that are separated by a matrix of urban areas are viewed as fragments.

The Metropolitan Natural Park features a surface area of 232 ha and is divided into seven defined management zones: primitive use, priority archaeological interest, intensive use, extensive use, special use, coordination of the Curundú River, and buffering influence [24]. The fragment under investigation in the PNM is located at 8°59′41.55″ N and 79°32′35.22″ W and features an approximate area of 18.12 ha alongside a perimeter of 1756 km. The vegetation observed at this site is characterized by a mixture of humid tropical forest and lowland tropical dry forest, and it features few areas of stubble and grasslands alongside a well-defined stratum; this region is used for hiking and is delimited by Juan Pablo Avenue. II, Avenida de la Amistad and the Metropolitan Equestrian Club. The fragment in Corozal (COR) is located at 8°59′19.34″ N and 79°34′11.83″ W and features an approximate area of 56.31 ha and a perimeter of 3028 km. The vegetation observed at this site can be characterized as herbaceous, including late secondary forests and some open areas. This forest fragment is managed by the Institutional Protection Service (SPI) and is used to support practice and training in the jungle. The fragment located in the town of Albrook (ALB) is located at 8°58′37.49″ N and 79°33′43.82″ W and features an approximate area of 34.79 ha and a perimeter of 5003 km. The vegetation observed at this site is heterogeneous and consists of open grasslands, stubble, and secondary forests. The site under investigation in this context is managed by the Panama Canal Authority; it is a heavily urbanized area in which many academic, commercial, and recreational activities take place. The fragment located in the City of Knowledge (CDS) is known as the Dr. Rodrigo Tarte Biological Reserve. This area is protected by the Ciudad del Saber Foundation. It is located at 9°00´3.13″ N and 79°35′12.96″ W. This site is adjacent to Soberanía National Park, and ex situ conservation activities are conducted at this location by the Ministry of the Environment (Table 1) (Figure 1).

These sites were selected from a qualitative perspective, which focused particularly on the degree of similarity among the biotic and physical characteristics of each site. The four sampling sites selected for this research are characterized by well-preserved secondary forests with similar types of vegetation; these locations have been fragmented by anthropogenic activities and pertain to the protection of the Panama Canal.

### 2.2. Sampling Method

Townes-type Malaise traps were used in this research as a passive trapping method, considering their ability to collect fast-moving flying insects [25,26]. Two traps were installed in each fragment, which were separated from each other by 0.5 km and located at least 0.2 km from the edge of the fragment to minimize the edge effect. The traps were placed in open areas within the forest to intercept flight corridors. Malaise traps were exclusively used due to their proven effectiveness in capturing flying insects, particularly Hymenoptera and Diptera, which were the focal groups of this study. Although complementary methods such as light traps or interception traps could have enriched the sampling, a standardized and replicable approach was prioritized, considering logistical limitations and the need to ensure comparability between sites in urban contexts. The sampling process took place weekly between August 2019 and March 2020 for a period of 33 weeks. The traps remained active continuously, thus generating a total effort of 5760 h of sampling and leading to the collection of 264 samples.

The samples thus collected were transported to the building of the Central American Program of Master’s in Entomology of the University of Panama (PCMENT). In one of the laboratories located in this building, a LEICA S9 stereoscope (Leica Microsystems, Wetzlar, Germany) was used to separate individuals of the families of interest, which were subsequently stored in vials with 95% ethanol. These individuals were subsequently mounted on entomological pins (1, 2, and 3), separated into morphospecies, and identified by reference to the relevant literature. The key provided in [27] was used to identify the subfamilies of Ichneumonidae Latreille, 1802. For Dolichopodidae Latreille, 1809, and Asilidae Meigen, 1802, we used the genera keys in [28]. For the genus Bethylidae Latreille, 1802, the key developed in [29,30] was used, and the illustrated key to the subfamilies and genera present in Panama was provided in [31]. For Psocoptera Schönherr, 1813, we used the keys developed in [32]. Alfredo Lanuza, a specialist in Coleoptera, identified individuals in the family Curculionidae Linnaeus, 1758. For the genus Membracidae Latreille, 1802, the works by [33] and Flynn (2012) [34] were used as a reference for identification.

In this study, the morphospecies approach proposed in [35] was employed. This approach facilitates the use of morphological groupings as operational units in the absence of complete taxonomic identification. This method also facilitates the estimation of richness and the analysis of species turnover among sites, provided that each morphospecies is recognized consistently based on visible diagnostic characteristics. With this system, it is possible to construct comparable sets of species and perform biodiversity assessments in habitat fragments, even in the absence of precise information regarding the taxonomic identities of all the organisms involved in this process. Oliver and Beattie demonstrated the efficacy of this approach in the context of studies on ants, beetles, and spiders, revealing that it is superior to traditional methods in terms of its ability to detect variations and predict ecological impacts in a more sensitive way.

### 2.3. Data Analysis

Statistical analyses were performed with the assistance of R version 4.2.2 software and the SpadeR platform. To determine the diversity and composition of the communities of the families under investigation, statistical analyses of alpha (α) diversity were conducted, including by reference to the Shannon-Wiener index, which can facilitate the integration of both species richness and equitability in terms of the distribution of individuals and thus represents a composite measure of community heterogeneity [36]. In turn, Simpson’s index can be used to estimate the probability that two randomly selected organisms belong to different species and is interpreted as a weighted average based on the relative abundance of the species present [37]. Similarly, the Pielou equity index was used to evaluate the degree of uniformity in the distribution of individuals among different species. To complement these analyses, accumulation and rarefaction curves were used, which can facilitate estimations of the expected wealth in communities that are homogeneous [38]. With respect to beta diversity, the Bray-Curtis [39] and Jaccard [40] indices and the Diserud-Odegaard multiple similarity index [41] were calculated. These metrics can facilitate the quantification of the turnover of species between communities or sites, thus reflecting the degree of biotic change along environmental or spatial gradients [42,43]. In addition, total species richness and true diversity were estimated via the SpadeR platform, and the abundance data pertaining to each site were used to compare the observed values with the expected values. On the other hand, Venn diagrams generated with the assistance of R (specifically, by reference to the matplotlib_venn package) were used to visualize the intersection of morphospecies among the different urban forest fragments, thus allowing for the number of exclusive and shared species to be identified and providing a graphic representation of the levels of community overlap among the sites.

## 3. Results

### 3.1. Diversity and General Abundance

A total of 2038 specimens that belonged to the seven families under investigation were collected alongside 43 subfamilies and 75 genera; furthermore, a total of 403 morphospecies were estimated at the four sites under investigation. The Ichneumonidae family exhibited a total of 841 individuals, followed by Curculionidae at 446 individuals, Dolichopodidae at 426 individuals, Bethylidae at 277 individuals, Asilidae at 108 individuals, Psocidae Schönherr, 1817 at 90 individuals, and Membracidae at 64 individuals. The Ichneumonidae family exhibited the greatest number of subfamilies, at a total of 18. The subfamilies that exhibited the highest numbers were Crypthorynchinae Schoenherr, 1825 (Curculionidae), Epyrinae Kieffer, 1914 (Bethylidae), and Smilinae Stal, 1869 (Membracidae), each of which was associated with six genera. These subfamilies were followed by Amphigerontiinae Pearman, 1936 (Psocidae), Baridinae Schoenherr, 1836, Curculioninae Latreille, 1802 (both from Curculionidae), and Membracinae Rafinesque, 1815 (Membracidae), which were associated with five genera each. On the other hand, the subfamilies Molytinae Schoenherr, 1823, and Conoderinae Schoenherr, 1833 (Curculionidae), alongside Pristocerinae Kieffer, 1914 (Bethylidae), were each associated with four genera.

In terms of abundance, the subfamily Sciapodinae Aldrich, 1905 (Dolichopodidae), was the most represented, at 389 samples, followed by Cryptinae Förster, 1869 (Ichneumonidae), at 243 individuals, Pristocerinae (Bethylidae) at 224, Conoderinae (Curculionidae) at 176, and Molytinae (Curculionidae) at 151 individuals. With respect to the number of morphospecies by genus, *Dissomphalus* Ashmead, 1893 (Bethylidae), was the most notable at 28 morphospecies, followed by *Zygops* Schoenherr, 1825 (Curculionidae), at 26, *Conotrachelus* Dejean, 1835 (Curculionidae), at 25, *Lechriops* Schoenherr, 1836 (Curculionidae), at 17, and *Anthonomus* Germar, 1817 (Curculionidae), at 7 morphospecies. In terms of abundance by genus, the most represented example was *Condylostylus* Bigot, 1859 (Dolichopodidae), at 277 specimens, followed by *Zygops* at 159, *Dissomphalus* at 151, *Conotrachelus* at 105, and *Hercostomus* Loew, 1857 (Dolichopodidae), at 104 individuals. Finally, the most abundant morphospecies identified during the sampling process were *Condylostylus* sp1 (Dolichopodidae) at 152 individuals, Cryptinae M1 (Ichneumonidae) at 127, *Hercostomus* sp1 at 126, *Zygops tridentata* Gyllenhal, 1836, at 102, and *Condylostylus* sp3 at 98.

The urban forest fragment that featured the highest abundance of specimens was Metropolitan Natural Park (PNM), at a total of 826 individuals, followed by Ciudad del Saber (CDS) at 676 individuals, Corozal (COR) at 280, and Albrook (ALB) at 256. In terms of the richness of the morphospecies, the fragment that featured the highest number was Ciudad del Saber, at 223 morphospecies, followed by PNM at 211, Corozal at 127, and Albrook, which exhibited the lowest level of richness at 118 morphospecies (Figure 2).

### 3.2. Alpha and Beta Diversity

The Shannon-Wiener (H′) diversity index values exceeded 3.0 for all fragments, thus indicating highly heterogeneous communities. The highest value was recorded in CDS (4.57), followed by both PNM and COR (4.4) and, at a slightly lower level, ALB (4.3). With respect to Margalef’s wealth index, the values ranged between 21.1 and 34.07. Ciudad del Saber exhibited the highest value (34.07), followed by PNM (31.27), Corozal (22.36), and Albrook (21.1) (Table 2) (Figure 3).

The values of Pielou’s evenness index (J), which ranged from 0.83 to 0.91, indicate a moderate to high level of equity among species. Although not perfectly even, these values suggest a relatively balanced distribution of abundances, with no single species dominating the communities, which is consistent with a moderately stable ecological structure [37] With respect to the Simpson dominance index (D), values close to zero were observed, thus suggesting that no morphospecies predominated significantly in the communities evaluated as part of this research. In addition, the high values of the inverse Simpson index (1-D), which were close to 1.0, reinforce the evidence indicating a high level of diversity and a balanced community structure in the urban forest fragments under investigation (Table 2).

With respect to beta diversity, the Bray–Curtis dissimilarity index indicated a high level of variation in the composition of morphospecies among the sites (0.7558). The greatest differences were observed between Corozal and Ciudad del Saber, whereas Ciudad del Saber and Parque Natural Metropolitano were the most similar to one another. The Jaccard similarity index confirmed these patterns: CDS and PNM were the most similar fragments (0.3111), followed by Corozal and Albrook (0.289). The lowest level of similarity was observed between CDS and COR (Figure 4). A permutational multivariate analysis of variance (PERMANOVA) based on Bray–Curtis dissimilarity was conducted to test for differences in insect community composition among urban forest fragments. The results indicated that the differences were not statistically significant (F = 0.9793, R^2^ = 0.27093, *p* = 0.395), suggesting that, based on the current sampling, there is no strong evidence of compositional divergence between fragments at the scale evaluated.

To complement the alpha and beta diversity indices, an estimate of the true richness and diversity of morphospecies was performed, which revealed marked differences among the urban fragments evaluated in this study. The Metropolitan Natural Park (PNM) and Ciudad del Saber (CDS) exhibited the highest abundances and levels of wealth observed in this context, at 211 and 223 morphospecies, respectively, which accounted for 51.88% and 48.13%, respectively, of the estimated wealth for each site. In contrast, the Corozal (COR) and Albrook (ALB) fragments exhibited a lower proportion of species than our estimations (36.70% and 42.8%, respectively), thus suggesting that a greater number of species were not detected during the sampling period. Despite these differences in richness, the true diversity observed (which was based on the distribution of abundances) was similar in all the fragments, as indicated by percentages that were higher than 95% with respect to the estimated values, thus indicating a relatively balanced community structure in terms of equity (Table 3).

The intersection analysis of species on the basis of Venn diagrams (A–D) allows us to visualize the distribution and coincidence of species among the urban forest fragments evaluated in this research: City of Knowledge (CDS), Metropolitan Natural Park (PNM), Corozal (COR) and Albrook (ALB). Diagram A (CDS–PNM–ALB) indicates a notable intersection between CDS and PNM, two fragments that feature higher levels of forest cover and closer geographic proximity, whereas ALB is characterized by a smaller number of shared species, thus suggesting a certain degree of ecological isolation. In diagram B (CDS–COR–ALB), CDS continues to exhibit the highest number of unique and shared species, and COR exhibits some moderate coincidences, possibly as a result of residual connectivity; ALB once again exhibits a low level of similarity with the other fragments. Diagram C (PNM–COR–ALB) reinforces this trend; namely, PNM appears as a middle point of intersection, whereas COR and ALB are characterized by a reduced coincidence of species. Finally, diagram D (CDS–PNM–COR) reveals the most ecologically relevant combination: CDS and PNM share a significant number of species, thus consolidating their role as nuclei of urban biodiversity, whereas COR is distinguished by a lower level of intersection, thus indicating its relative isolation (Figure 5).

Finally, the Diserud–Odegaard similarity index for all fragments is 0.543 (54%), thus indicating a moderate level of similarity in terms of the composition of the corresponding morphospecies. The multiple dissimilarity index estimated between the fragments is 0.407, thereby reflecting notable differences in the structure of the evaluated communities. We calculated alpha diversity indices (Shannon, Simpson, and Chao1) for each forest fragment and then compared them using ANOVA and Kruskal–Wallis tests. While all fragments showed high levels of diversity (Shannon > 4.3; Simpson > 0.97), statistical tests did not detect significant differences among sites (Kruskal–Wallis χ^2^ = 3.00, df = 3, *p* = 0.3916) (Table 4). However, a PERMANOVA based on Bray–Curtis distances revealed significant differences in species composition among fragments (R^2^ = 0.0389, F = 2.91, *p* = 0.001), suggesting that although overall diversity levels were similar, each fragment harbored distinct insect communities. This pattern was also supported by the normalized Morisita–Horn similarity index (NDMI), which showed higher compositional similarity between Ciudad del Saber and Parque Natural Metropolitano and lower similarity values between Corozal and the other fragments (Figure 6).

## 4. Discussion

The results obtained in this research regarding the diversity and composition of seven families of insects in urban forest fragments near Panama City highlight a similarity between communities (Diserud–Odegaard = 0.543), thus contradicting the initial hypothesis that a high level of similarity is to the result of the geographic proximity and shared ecological characteristics among the fragments. This divergence suggests that other factors that were not measured directly in this study could modulate the community structure observed in this context.

Our findings agree with those reported in [44], who demonstrated that the relationship between environmental heterogeneity and diversity is not always linear or predictable, even with respect to similar ecological groups. This finding reinforces the claim that biological communities can respond differently to environmental pressures, even under apparently homogeneous conditions.

In contrast, our results differ from those that have been reported in studies conducted in other contexts, such as [45], which focused on Sydney, Australia, where wasp communities exhibit high levels of similarity among fragments, thus suggesting some degree of resistance to urban fragmentation. These findings also differ from the results of previous investigations conducted in Panama, such as the studies conducted by [46], who focused on butterflies, and by [1], who focused on wasps of the Braconidae family, which have revealed highly similar compositions among fragments (as indicated by dissimilarity rates of 3% and 6%, respectively). This discrepancy could be the result of the fact that a greater number of families were included in our study, thus increasing the level of variability among samples or methodological differences pertaining to the calculation of dissimilarity.

At the taxonomic level, the subfamilies Crypthorynchinae (Curculionidae), Pristocerinae (Bethylidae), and Smilinae (Membracidae) exhibit the greatest number of genera, a finding which is similar to the results of previous studies that have highlighted the high levels of diversity observed in this context in tropical regions [47,48,49,50]. With respect to the richness of the morphospecies, the fragments associated with the City of Knowledge and the Metropolitan Natural Park were associated with the highest values, possibly as a result of their greater connectivity, lower levels of isolation, or status as protected areas. In contrast, Corozal and Albrook, which are more firmly embedded in an urban matrix, exhibited less wealth and abundance. This trend was also documented by [51], who noted that the size of the fragment and its connectivity with other forest masses significantly influence the population dynamics of the corresponding organisms.

Another possible factor contributing to the variation in community richness and abundance across fragments is the edge effect. Although Malaise traps were placed approximately 200 m from the visible boundaries of each fragment to minimize edge influence, this distance may not have been sufficient, particularly in small or irregularly shaped forest patches. Previous research has shown that edge effects in tropical forests can extend beyond 300 m [52], and such effects may be amplified in urban landscapes due to structural fragmentation and anthropogenic disturbance. This methodological limitation should be considered when interpreting the observed patterns.

The dominance of parasitoid families, particularly Ichneumonidae and Bethylidae, supports the idea that urban forest fragments provide suitable habitats for complex trophic interactions. These families include numerous species that act as natural enemies of phytophagous and detritivorous insects, playing a critical role in regulating pest populations and maintaining ecological balance. Their presence can enhance the resilience of fragmented ecosystems by supporting top–down control mechanisms and contributing to the stability of food web dynamics in urban environments. Their presence is often associated with the availability of suitable hosts in urban environments, as observed for braconids in Panama [1]. The high values of alpha diversity indices (Shannon, Margalef, Simpson, and inverse Simpson) observed across all fragments suggest the presence of rich and relatively equitable insect communities. This lack of dominance can be interpreted as an indicator of ecological stability and the potential use of these fragments as biological corridors.

These results emphasize the functional role of urban forest remnants as reservoirs of entomological biodiversity and key components of broader ecological networks, particularly in tropical cities. Their ability to sustain diverse insect communities is influenced by fragment size, habitat quality, and especially the degree of connectivity with other green spaces [53]. Well-connected fragments have been shown to facilitate gene flow, promote species dispersal, and mitigate the negative impacts of habitat isolation, including biodiversity loss and reduced ecological interactions [54].

Observed species richness values—especially when contrasted with estimated richness—indicate that a portion of the community remains undetected, possibly due to cryptic behaviors, seasonality, or sampling limitations [55,56]. Although the original sampling design intended to cover both the rainy and dry seasons over a continuous 13-month period, fieldwork was interrupted in March 2020 due to the COVID-19 pandemic and strict national lockdowns in Panama. As a result, dry-season data were not collected, which may have limited the ability to fully capture seasonal variation in insect communities, particularly for taxa with strong phenological shifts. Fragments like CDS and PNM stood out as biodiversity hotspots, with high richness and many shared species, likely due to their larger size and better landscape connectivity [17,54]. The Venn diagrams used in this study visually demonstrate the overlap in species composition between fragments. Particularly, CDS and PNM share a notable proportion of morphospecies. Although exact percentages were not statistically tested for significance, the number of shared morphospecies between these two sites represents over 50% of the observed richness in each fragment, supporting the interpretation that they function as biodiversity nodes within the urban matrix. This overlap reflects their larger size and greater habitat continuity, as previously described. In contrast, more isolated fragments such as ALB and COR exhibited lower richness and more distinct communities, suggesting higher anthropogenic pressure and ecological fragmentation [18,53]. Although species richness and abundance varied among forest fragments, the PERMANOVA results showed no statistically significant differences in community composition (*p* = 0.395). This may be attributed to the relatively small number of traps per site, high internal variability, or the ecological similarity of the sites due to shared urban pressures. The NMDS plot, however, suggested some level of visual separation, indicating that a more intensive sampling design or inclusion of environmental covariates may better capture the functional differences across sites.

Given these findings, we recommend that urban planning integrates a landscape perspective that prioritizes not only the extent but also the spatial configuration and connectivity of vegetation. Strategies such as ecological restoration, enhancing structural connectivity, and regular seasonal monitoring of insect biodiversity should be considered essential components of sustainability plans aimed at conserving entomofauna in tropical urban environments [19,54].

The structure of the intersections observed among the sites highlights the importance of conserving and improving the connectivity among fragments such as CDS and PNM, as this situation can facilitate the movement of species, maintain gene flow, and promote more stable and resilient communities [19]. Similarly, the differences observed among the fragments highlight the need to implement ecological restoration measures in locations such as Corozal and Albrook, with the aim of increasing their ability to serve as urban ecological habitats or corridors. This strategy, in addition to efforts to strengthen local biodiversity, can contribute to the ecological sustainability of the urban landscape; accordingly, this approach is in line with contemporary approaches that involve green infrastructure and resilient planning to combat climate change [54,57].

The lack of statistically significant differences in alpha diversity indices among forest fragments suggests a relatively homogeneous level of insect diversity across urban patches. This result aligns with studies indicating that small but structurally complex urban green spaces can support high insect diversity regardless of size or isolation [9,58]. However, the significant differences found in community composition (PERMANOVA) and the contrasting similarity values in the NDMI matrix confirm that each fragment hosts a distinct assemblage of species. Similar patterns have been observed in other tropical urban systems, where beta diversity is driven more by habitat heterogeneity and microclimatic variation than by alpha diversity metrics alone [59,60]. These findings reinforce the importance of preserving multiple forest fragments within urban landscapes, as each contributes uniquely to regional biodiversity.

## 5. Conclusions

This study provides robust evidence of high insect diversity within urban forest fragments in Panama City, even in landscapes heavily altered by urbanization. A total of 2038 insect specimens were collected and classified into 403 morphospecies, representing 75 genera across seven families. The high richness, evenness, and absence of dominance indicate that these forest patches support structurally complex and ecologically balanced entomological communities.

Although a high level of similarity in species composition was expected due to the proximity and shared environmental conditions among the sites, the observed heterogeneity suggests that additional, unmeasured variables—such as microclimatic differences, habitat quality, or fragment history—may be influencing community structure.

True diversity and richness estimates revealed that while not all expected diversity was captured, the most common species were well represented, and the internal composition of each community was stable. Venn diagrams confirmed that key fragments such as the City of Knowledge (CDS) and Metropolitan Natural Park (PNM) share a greater number of species, suggesting higher ecological continuity. In contrast, Corozal and Albrook exhibited fewer shared species, likely reflecting their greater isolation and exposure to urban pressures. These analyses allowed us to identify both biodiversity nodes and critical areas for ecological restoration.

Differences in richness and abundance across sites may relate to factors such as the degree of isolation, fragment size, and the characteristics of the surrounding matrix. Future research should integrate landscape ecology tools, including connectivity and habitat quality metrics, to explore these patterns further.

Based on our findings, we recommend the following actions to strengthen biodiversity conservation in urban environments: Prioritize the ecological restoration of highly degraded fragments (e.g., Corozal and Albrook), including reforestation and habitat enrichment efforts, increase structural and functional connectivity among forest patches by implementing green corridors or stepping-stone vegetation zones, incorporate seasonal and long-term insect monitoring programs as bioindicators of ecological change, and integrate forest fragments into official urban development plans as critical components of green infrastructure to enhance urban resilience and ecosystem services.

Urban forest fragments should not be seen as isolated remnants but as integral parts of sustainable and resilient tropical cities. Their active conservation and management can contribute significantly to mitigating biodiversity loss and the ecological impacts of urban expansion in the tropics.

## Figures and Tables

**Figure 1 biology-14-00721-f001:**
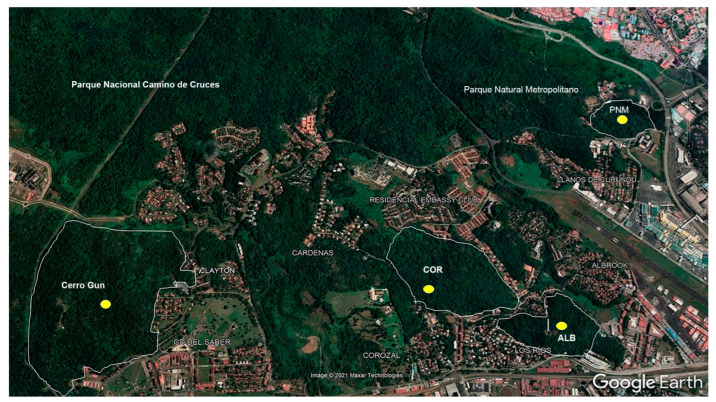
Sampling sites in the urban forests associated with Panama City.

**Figure 2 biology-14-00721-f002:**
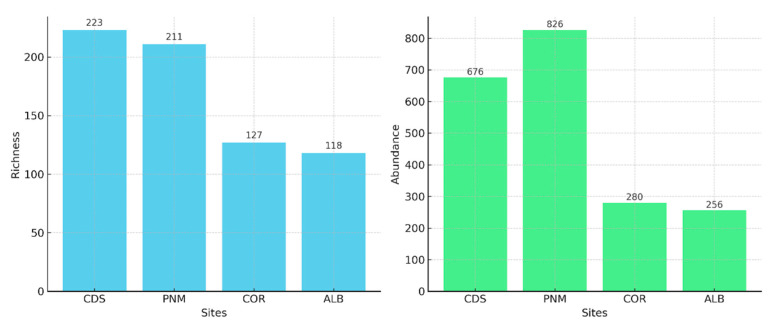
Number of individuals and morphospecies estimated in each of the four urban forest fragments sampled in Panama City.

**Figure 3 biology-14-00721-f003:**
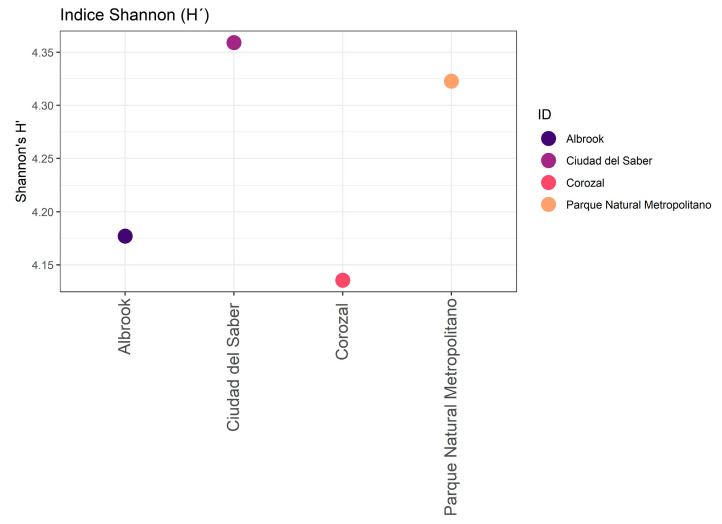
Shannon index (H′), in which context the diversity of the species in the urban forest fragments studied is observed.

**Figure 4 biology-14-00721-f004:**
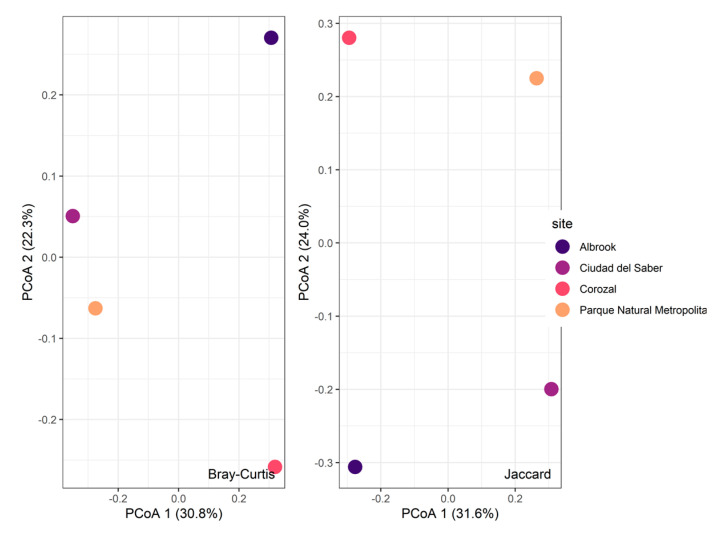
Bray–Curtis and Jaccard similarity indices of the morphospecies of the families under investigation in the different study fragments.

**Figure 5 biology-14-00721-f005:**
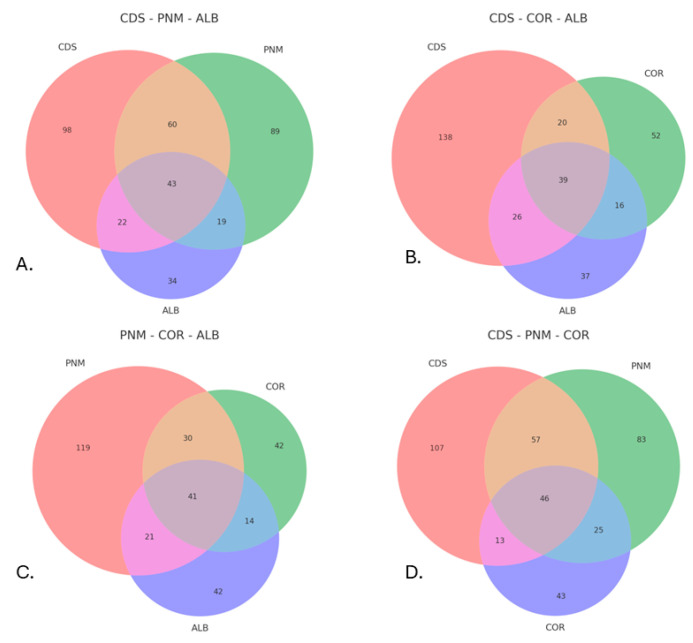
Venn diagrams that illustrate the intersections of insect morphospecies among three urban forest fragments in different combinations. The Ciudad del Saber (CDS), Parque Natural Metropolitano (PNM), Corozal (COR), and Albrook (ALB) sites are compared. (**A**) Intersections among CDS, PNM, and ALB; (**B**) CDS, COR, and ALB; (**C**) PNM, COR, and ALB; and (**D**) CDS, PNM, and COR.

**Figure 6 biology-14-00721-f006:**
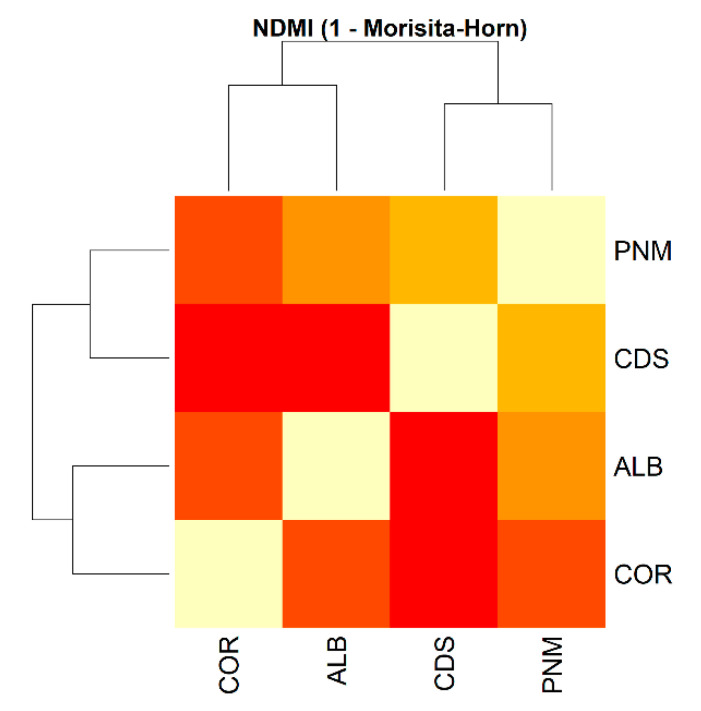
Heatmap showing the pairwise similarity between insect communities in four urban forest fragments in Panama City, based on the normalized Morisita–Horn index (NDMI). Higher values (close to 1) indicate greater similarity in species composition, while lower values reflect higher dissimilarity. The highest similarity was observed between Ciudad del Saber (CDS) and Parque Natural Metropolitano (PNM), while Corozal (COR) exhibited the greatest dissimilarity relative to the other fragments.

**Table 1 biology-14-00721-t001:** Study sites in Panama City, Panama.

Sites	Geographical Location	Altitude (Meters)	Vegetation	Annual Temperature (°C)
PNM	8°59′41.55″ N–79°32′35.22″ O	0–150	Tropical dry forest	28
CDS	9°00′24.3″ N–79°35′05.2″ W	0–100	Tropical dry forest	27.5
COR	08°59′19.34″ N–079°34′11.83″ W	30–60	Overlapping deciduous forest.	26.5
ALB	08°58′37.49″ N–079°33′43.82″ W	20–80	Overlapping deciduous forest.	26.5

**Table 2 biology-14-00721-t002:** Alpha (α) diversity indices of the urban forest fragments under investigation.

	Albrook	Ciudad del Saber	Corozal	Parque Natural Metropolitano
Taxas	118	223	127	211
Individuos	256	676	280	826
Dominancia (D)	0.02515	0.02836	0.01946	0.02785
Simpson (1-D)	0.9749	0.9716	0.9805	0.9721
Shannon (H′)	4.302	4.575	4.4	4.434
Margalef	21.1	34.07	22.36	31.27
Equidad (J)	0.9018	0.8461	0.9083	0.8285

**Table 3 biology-14-00721-t003:** Observed, estimated, and exclusive richness, alongside observed and estimated true diversity in Panama (the percentage of richness and diversity obtained in the field with respect to the estimations is shown in parentheses).

Sites	Abundance	True Diversity			
		Obs.	Est.	Obs.	Est.
PNM	826	211	406.7 (51.88)	103.307	106.810 (96.72)
CDS	676	223	463.3 (48.13)	126.970	133.482 (95.12)
COR	280	127	346.0 (36.70)	119.806	118.624 (1.00)
ALB	256	118	275.5 (42.8%)	108.86	111.97 (97.22%)

**Table 4 biology-14-00721-t004:** Alpha diversity indices of insect communities across four urban forest fragments in Panama City. Values are shown for Shannon diversity, Simpson dominance, and Chao1 estimated richness. The final row presents the mean and standard deviation (SD) across all fragments. Although diversity was consistently high across sites, a Kruskal–Wallis test found no statistically significant differences among fragments for the Shannon index (χ^2^ = 3.00, df = 3, *p* = 0.3916).

Sites	Shannon	Simpson	Chao1
Albrook	4.30	0.975	256.8
Ciudad del Saber	4.58	0.972	421.2
Corozal	4.40	0.981	320.7
Parque Natural Metropolitano	4.43	0.972	422.5
Mean ± SD	4.43 ± 0.12	0.975 ± 0.004	355.3 ± 78.9

## Data Availability

The data presented in this study are available on request from the corresponding author.

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
