# Peer review of "The Diversity and Composition of Insect Communities in Urban Forest Fragments near Panama City"

_biology, 2025, doi:10.3390/biology14060721_

Round 1

Reviewer 1 Report

Comments and Suggestions for Authors

The study provides valuable data on urban insect diversity but requires methodological rigor (seasonal sampling, edge-effect analysis) and statistical depth (multivariate tests) to support its conclusions. Taxonomic consistency and language refinement are also needed.

  1. The traps were placed "0.2 km from the edge" to minimize edge effects, but no justification or citation supports this distance. Edge effects in tropical forests can extend >100–300 m (Laurance et al. 2002), and fragments like ALB (perimeter 5,003 km) likely have pronounced edges. 
  2. Introduction needs to be more updated with some latest studies on insect populations and there dynamics. Refer to these useful papers, https://doi.org/10.1016/j.landurbplan.2021.104238

https://doi.org/10.1016/j.jnc.2024.126602

https://doi.org/10.1007/s10980-023-01747-2

https://doi.org/10.1016/j.scitotenv.2025.179660

  1. Sampling occurred from August 2019–March 2020, covering the rainy season but omitting the dry season (December–April). This neglects seasonal variations in insect activity, potentially skewing diversity estimates
  2. Beta diversity is assessed via indices (Bray-Curtis/Jaccard), but no multivariate analysis (e.g., NMDS, PERMANOVA) is used to test if compositional differences are statistically significant 

5. Subfamilies spelled variably: "Smilinae" (p. 5) vs. "Smiliniae" (p. 11); "Epyrinae" (p. 5) vs. "Epyrinae" (p. 16, Ref 45).

6. Families: "Bethyliidae" (Abstract) should be Bethylidae

  1. Pielou's equity (J) values (Table 2) range from 0.83–0.91, yet the text states they are "close to 1.0" (p. 7). Values <0.9 indicate moderate unevenness (Magurran 2004).
  2. Qualitative descriptions of species overlap (e.g., "CDS and PNM share significant species") are not quantifiedwith shared-species metrics or tested for significance.

9. "Indice Shannon" → "Índice Shannon"; "Cruidad del Saber" → "Ciudad del Saber".

Author Response

Please find our responses in the attachment

Reviewer 2 Report

Comments and Suggestions for Authors

The authors have addressed the important topic of the role of urban forest fragments in maintaining insect diversity. In the face of rapid urbanisation in the tropics, such research is important for planning sustainable land use and biodiversity conservation. The work makes a valuable contribution to the understanding of the extent to which transformed habitats can continue to act as biodiversity reservoirs.Up to 2,038 insect specimens were collected, classified into 403 morphospecia, 75 genera and 43 subfamilies. The morphospecia-based methodology is widely used in tropical entomology, where the taxonomy of many groups is still poorly understood. The paper provides strong evidence that urban forests, even if they are only fragments of primary ecosystems, can act as important refugia for insects.

Introduction

The introduction is based on a solid foundation of literature and current ecological issues.  The objectives are clearly defined and follow directly from the research problems presented. Although the excerpt from the introduction announces the diversity analysis, it would be useful to elaborate on why these seven families were chosen - are they indicative, well described or easy to identify morphologically?

Methods

Why did the authors only use Malaise traps? One could briefly justify the abandonment of complementary methods to show an awareness of the limitations of this one technique.

Results

I have no complaints about the results.

Discussion
Lines 347-377 and 379-397 could be reduced to a single, coherent section on the functional role of fragments and recommendations for urban planning.

Ichneumonidae and Bethylidae are important from a biological control point of view - this was worth elaborating on: how might their presence affect ecosystem stability?

Conclusions
A synthetic summary of the quantitative results is missing here. The text says that fragments “should be part of sustainability plans”, but does not indicate what actions should be prioritised: e.g. restoration, increasing connectivity, seasonal monitoring? The authors can add clear recommendations here.

References

In records such as [5] and [13] there is a “-” character in the DOI, which divides the number into two lines.

Book and chapter citations (e.g. [2], [17], [27], [32]) are not always consistent. It would be good to standardise the format. Once an article has been accepted by the editors, this should be standardised.

Author Response

Dear reviewer, 

We sincerely thank reviewer for their thoughtful and constructive comments, which greatly helped improve the clarity, coherence, and scientific rigor of our manuscript. All suggestions have been carefully addressed in the revised version, including:

  • Justification for the selection of the seven insect families in the Introduction
  • Clarification of the exclusive use of Malaise traps in the Methods section
  • Expansion of the ecological role of Ichneumonidae and Bethylidae in the Discussion
  • Reorganization and unification of the discussion on functional roles and urban planning recommendations
  • Inclusion of a concise quantitative summary and clear recommendations in the Conclusions
  • Corrections to references, including formatting of DOIs and standardization of book and chapter citations.

We hope that these changes meet your expectations and strengthen the overall quality and impact of the manuscript.

Round 2

Reviewer 1 Report

Comments and Suggestions for Authors

I am satisfied with the revised manuscript. However, authors are requested to prepare the response report in standard format in future and quote the actual reviewer comments rather than mentioning only a part of the reviewer's comments.